# Training-Free Zero-Shot Anomaly Detection in 3D Brain MRI with 2D Foundation Models

**Tai Le Gia** [ID]                   GIATAILE@O.CNU.AC.KR
*Department of Mathematics, Chungnam National University, South Korea*

**Jaehyun Ahn**                    JHAHN@CNU.AC.KR
*Department of Mathematics, Chungnam National University, South Korea*

**Editors:** Accepted for publication at MIDL 2026

## Abstract

Zero-shot anomaly detection (ZSAD) has gained increasing attention in medical imaging as a way to identify abnormalities without task-specific supervision, but most advances remain limited to 2D datasets. Extending ZSAD to 3D medical images has proven challenging, with existing methods relying on slice-wise features and vision–language models, which fail to capture volumetric structure. In this paper, we introduce a fully training-free framework for ZSAD in 3D brain MRI that constructs localized volumetric tokens by aggregating multi-axis slices processed by 2D foundation models. These 3D patch tokens restore cubic spatial context and integrate directly with distance-based, batch-level anomaly detection pipelines. The framework provides compact 3D representations that are practical to compute on standard GPUs and require no fine-tuning, prompts, or supervision. Our results show that training-free, batch-based ZSAD can be effectively extended from 2D encoders to full 3D MRI volumes, offering a simple and robust approach for volumetric anomaly detection. The code used in this study is available at https://github.com/DumBringer/CoDeGraph3D.

**Keywords:** magnetic resonance imaging, zero-shot anomaly detection, 3D volumetric representation, image-level detection, voxel-level segmentation

## 1. Introduction

Anomaly detection is essential in medical imaging, where early identification of abnormal structures supports diagnosis and treatment planning. Conventional unsupervised anomaly detection (UAD) methods rely on large sets of clean, domain-specific training data, which are costly to obtain for volumetric medical imaging. Zero-shot anomaly detection (ZSAD) offers an appealing alternative by removing the need for supervised training, yet most advances remain limited to 2D images (Jeong et al., 2023; Zhou et al., 2023; Chen et al., 2023; Li et al., 2024; Gia and Ahn, 2025). Extending ZSAD to 3D MRI volumes is nontrivial: there are no 3D foundation models, and simple slice-wise features cannot capture full volumetric structure. Recent CLIP-based attempts (Marzullo et al., 2025) highlight this difficulty, showing that naive extensions of 2D ZSAD pipelines to 3D yield unstable performance.

Existing ZSAD approaches for 2D images follow two main directions. Text-based methods (Jeong et al., 2023; Zhou et al., 2023; Chen et al., 2023) use vision–language models to score abnormalities via text prompts, but often require prompt tuning or additional training to achieve satisfactory performance. Batch-based methods (Li et al., 2024; Gia and Ahn, 2025) instead operate purely on visual tokens extracted by vision transformers

and exploit the intrinsic structure and statistics of these tokens across a batch of images. These methods exploit the statistical observations: normal patches consistently find similar counterparts in other images, whereas anomalous patches are rare and distinctive. By performing cross-sample similarity searches, batch-based approaches isolate such outliers without any prompts or supervision. However, extending this paradigm directly to 3D MRI is not straightforward. First, there are currently no general-purpose foundation models (akin to DINOv2 or CLIP) for volumetric data. Second, volumetric images generate orders of magnitude more tokens than 2D images, so naive tokenization causes extreme memory demands and renders mutual similarity computations computationally intractable.

In this paper, we propose a training-free batch-based ZSAD framework for 3D brain MRI to address these limitations. Our approach constructs localized volumetric tokens by aggregating multi-axis 2D foundation-model features into cubic 3D patches. This strategy preserves essential 3D spatial context while drastically reducing the number of tokens per volume, bringing the token count back into a regime where batch-based methods are feasible. To further reduce computational and memory burden, we apply a random projection to the token features, compressing their dimensionality while approximately preserving neighborhood geometry. The resulting compact 3D tokens can then be processed with standard batch-based approaches such as MuSc (Li et al., 2024) and CoDeGraph (Gia and Ahn, 2025), without any fine-tuning, prompts, or task-specific supervision.

Our main contributions are:

- We introduce the first practical batch-based ZSAD framework for 3D brain MRI, extending fully training-free principles from 2D to volumetric data.

- We propose a multi-axis volumetric tokenization and random projection pipeline that preserves cubic spatial context while enabling tractable mutual similarity computations for 3D volumes.

- Through extensive experiments, we show that our method outperforms representative CLIP-based ZSAD baselines and, in some cases, matches or exceeds the performance of supervised methods.

## 2. Related Work

**Anomaly Detection in Brain MRI.** Most unsupervised anomaly detection methods for brain MRI rely on reconstruction-based models—such as Autoencoders (Atlason et al., 2019; Baur et al., 2021; Cai et al., 2024), VQ-VAE (Pinaya et al., 2022b), GANs (Schlegl et al., 2019), or diffusion models (Pinaya et al., 2022a; Wu et al., 2024)—that must be trained on large collections of normal 3D MRI volumes to learn a representation of healthy anatomy. These approaches operate on the assumption that a model trained exclusively on healthy data will fail to accurately reconstruct unseen pathological features, allowing anomalies to be segmented via the residual error. While effective in controlled settings, their behavior depends heavily on the specific distribution of the training set. Consequently, these models are often brittle when deployed across different scanners or acquisition protocols.

**Zero-Shot Anomaly Detection.** To eliminate the need for training data, Zero-Shot Anomaly Detection (ZSAD) leverages representations from pre-trained foundation mod-

els. The dominant approach aligns visual features with textual descriptions of normality and pathology using vision–language models like CLIP (Jeong et al., 2023; Zhou et al., 2023; Chen et al., 2023). However, applying this strategy to medical imaging is problematic due to the significant domain gap and the difficulty of crafting robust clinical text prompts (Marzullo et al., 2025). A parallel emerging paradigm, *batch-based* ZSAD (Li et al., 2024; Gia and Ahn, 2025), discards language entirely; instead, it identifies anomalies as statistical outliers within a batch of visual tokens extracted by pure vision encoders (e.g., DINOv2). While effective for 2D industrial inspection, the validity of this batch-centric paradigm for volumetric medical data remains unproven. To the best of our knowledge, this work is the first to demonstrate that batch-based principles can be effectively extended to 3D medical imaging, establishing a viable, training-free path for volumetric anomaly detection.

**3D Representation Learning.** The development of native 3D foundation models is constrained by the scarcity of large-scale volumetric datasets and the high computational cost of training 3D networks. Consequently, recent work has explored adapting pre-trained 2D encoders to 3D data without additional training. A notable example is RAPTOR (An et al., 2025), which constructs scalable 3D volume-level embeddings by aggregating 2D slice features across multiple axes. Building on this training-free paradigm, we extend multi-axis aggregation to the *local* level: rather than producing a single global descriptor, we generate dense and spatially coherent 3D patch tokens. This design preserves the granular spatial context required for voxel-wise anomaly segmentation.

## 3. Method

### 3.1. Batch-Based Anomaly Detection on Collections of Tokens

Let the test dataset be $\mathcal{B} = \{C_1, \ldots, C_B\}$, where each collection $C_i$ consists of a finite set of feature tokens $C_i = \{\mathbf{z}_i^1, \ldots, \mathbf{z}_i^N\} \subset \mathbb{R}^D$, typically extracted using foundation models such as DINOv2 (Oquab et al., 2023). For a query token $\mathbf{z} \in C_i$, its distance to another collection $C_j$ ($j \neq i$) is defined as the nearest-token distance:

$$d(\mathbf{z}, C_j) = \min_k \|\mathbf{z} - \mathbf{z}_j^k\|_2.$$

Sorting these distances across all other collections yields the *Mutual Similarity Vector* (MSV):

$$\mathcal{D}_{\mathcal{B}}(\mathbf{z}) = [\, d(\mathbf{z})_{(1)}, \ldots, d(\mathbf{z})_{(B-1)} \,],$$

where $d(\mathbf{z})_{(t)}$ is the $t$-th smallest cross-collection distance.

Batch-based methods rest on the *Doppelgänger assumption*: normal structures tend to recur across different samples, whereas abnormal or rare patterns do not. Consequently, normal tokens typically find several close matches in the dataset (resulting in small initial MSV values), while anomalous tokens have fewer such matches and thus higher MSV values (resulting in large MSV values). MuSc (Li et al., 2024) turns this observation into an anomaly score by averaging the first $K$ entries of the MSV:

$$a(\mathbf{z}) = \frac{1}{K} \sum_{t=1}^{K} d(\mathbf{z})_{(t)}.$$

In practice, $K$ is set to a small fraction of the dataset size (typically 10–30% of $B$), which improves robustness to noise and rare normal patterns. MuSc further aggregates scores from multiple layers and receptive fields, enhancing detection of anomalies at different scales. A collection-level anomaly score is derived by taking the maximum token-level score.

A fundamental challenge in batch-based ZSAD is the presence of *consistent anomalies*: similar anomalies that repeat across multiple collections can make anomalous tokens become mutual nearest neighbors, artificially lowering their scores. CoDeGraph (Gia and Ahn, 2025) addresses this by detecting collections that share consistent-anomaly patterns and selectively excluding the suspicious tokens from MSV computation, thereby restoring the validity of rarity-based scoring.

Crucially, this entire pipeline operates on unordered sets of tokens extracted from test samples, and requires no task-specific training or text prompts. However, applying this logic to 3D data requires a mechanism to transform high-dimensional volumetric samples $V_i$ into discrete, semantically rich token sets $C_i$. Section 3.2 introduces our training-free 3D patch extraction method designed to bridge this gap.

### 3.2. Multi-Axis 3D-Patch Tokenization

To bridge the gap between continuous volumetric data and the discrete token requirements of the batch-based framework (Sec. 3.1), we introduce a training-free tokenization pipeline. This process leverages frozen 2D foundation models to extract semantic features while restoring the 3D spatial coherence lost in standard slice-wise approaches. We note that our design is encoder-agnostic: any 2D vision transformer can be used without altering the pipeline (see Appendix A).

#### 3.2.1. Axis-wise Extraction and Patch-Aligned Pooling

Let a preprocessed MRI volume be a cubic grid $V \in \mathbb{R}^{H \times H \times H}$. We decompose $V$ along the three anatomical axes—axial, coronal, and sagittal. For a given axis (e.g., axial), the volume is treated as a sequence of $H$ slices. Each slice $S_h$ is processed by a frozen 2D encoder $f(\cdot)$ (e.g., DINOv2) with patch size $p$. This yields a feature grid for each slice:

$$S_h \xrightarrow{f} \left\{ \mathbf{f}_{(h,u,v)} \in \mathbb{R}^D : 1 \leq u, v \leq N_p \right\},$$

where $N_p = H/p$ is the grid resolution and $D$ is the feature dimension. Stacking these outputs results in a tensor of size $H \times N_p \times N_p \times D$. Directly using this representation is computationally intractable for batch pairwise comparisons (e.g., $\approx 58$ million floating points per axis for a $224^3$ volume).

To enforce computational efficiency and restore volumetric cubicity, we apply *patch-aligned average pooling*. We group the $H$ slices into non-overlapping blocks of depth $p$, matching the encoder's native patch resolution. For a target 3D coordinate $(x, y, z)$ in the downsampled grid $N_p \times N_p \times N_p$, the feature is aggregated over the corresponding block of slices $\mathcal{G}_x$:

$$\mathbf{z}^{(\text{axis})}_{(x,y,z)} = \frac{1}{p} \sum_{h \in \mathcal{G}_x} \mathbf{f}^{(\text{axis})}_{(h,y,z)}, \quad \text{followed by } \ell_2\text{-normalization.} \tag{1}$$

This operation effectively downsamples the spatial resolution by a factor of $p$ in the slicing dimension, resulting in a cubic token representing a $p \times p \times p$ voxel region. This process is repeated for all three axes, permuting coordinates to align them to a unified $(x, y, z)$ grid.

### 3.2.2. RANDOM PROJECTION AND MULTI-VIEW FUSION

To further reduce computational cost while preserving geometric structure for batch-based methods, we apply random projection based on the Johnson–Lindenstrauss lemma (Johnson et al., 1984). The lemma ensures that pairwise distances are approximately preserved under low-dimensional embeddings, making random projection well suited to the distance-based nature of our anomaly scoring. Hence, we apply a fixed Gaussian random matrix $\mathbf{R} \in \mathbb{R}^{D \times k}$ with $k \ll D$ (e.g., $k = 128$) to project the tokens from each axis:

$$\mathbf{v}^{(\text{axis})}_{(x,y,z)} = \mathbf{R}^{\top} \mathbf{z}^{(\text{axis})}_{(x,y,z)} \in \mathbb{R}^{k}. \tag{2}$$

Finally, to integrate complementary anatomical context, we concatenate the projected features from the axial, coronal, and sagittal views at each spatial location:

$$\mathbf{v}_{(x,y,z)} = \left[ \mathbf{v}^{(\text{ax})}_{(x,y,z)}, \mathbf{v}^{(\text{cor})}_{(x,y,z)}, \mathbf{v}^{(\text{sag})}_{(x,y,z)} \right] \in \mathbb{R}^{3k}. \tag{3}$$

Flattening this grid yields the final collection $C_i = \{\mathbf{v}^1_i, \ldots, \mathbf{v}^N_i\}$ of volume $V_i$, where $N = N_p^3$ and each $\mathbf{v}^k_i$ represents a 3D patch at $(x, y, z)$ in (3). This collection is compact, semantically rich, and spatially localized, satisfying the input requirements for the batch-based anomaly detection described in Sec. 3.1.

**Background Suppression.** Standard preprocessing (e.g., skull-stripping) leaves large regions of zero-valued background voxels. Including these "void" tokens is computationally wasteful and introduces artificial redundancy that can distort batch statistics. We therefore utilize the binary brain mask to filter out background tokens prior to processing. This step significantly reduces the computational load of MSV calculations and ensures that the similarity graph constructed in CoDeGraph (Gia and Ahn, 2025) is driven solely by biological tissue patterns rather than background correlations.

**Framework Integration.** Once each MRI volume is transformed into a token collection $C_i = \{\mathbf{v}^1_i, \ldots, \mathbf{v}^N_i\}$, batch-based anomaly detection (Section 3.1) is applied directly to these sets. Running MuSc or CoDeGraph on $C_i$ produces a volumetric anomaly score map of size $N_p \times N_p \times N_p$, aligned with the cubic grid of 3D patch tokens. This coarse anomaly map is then trilinearly resized to the original resolution $H \times H \times H$ to obtain voxel-wise scores. Background voxels that were excluded from MSV computation retain an anomaly score of zero in the final output.

## 4. Experiments

### 4.1. Experimental Setup

**Datasets and Preprocessing** We evaluate the framework on volumetric Anomaly Classification (AC) and Anomaly Segmentation (AS) using IXI (IXI) (healthy) and BraTS-2025 METS (Maleki et al., 2025) (tumor). Both T2-weighted and native T1-weighted scans were

used. The datasets are split into an 80% portion used solely for supervised baselines and a 20% portion reserved for evaluation. The final mixed test batch comprises 180 volumes (115 IXI, 65 BraTS), and inference is performed jointly across the full batch.

All volumes undergo a standardized preprocessing pipeline. Each scan is first registered to the SRI-24 atlas (Rohlfing et al., 2010) using CaPTk (Pati et al., 2019), then skull-stripped using HD-BET (Isensee et al., 2019). Since skull-stripping produces substantial empty borders, the central brain region is cropped to a cube of $156^3$ voxels to remove most background. The cropped volume is then resampled to $224^3$, histogram-standardized (Nyúl et al., 2000) using a fixed reference template, and normalized to $[0, 1]$. For BraTS segmentations, we follow the common binary setup and treat all voxels with labels greater than 0 as anomalies.

**Implementation Details**   Tokenization follows the multi-axis procedure described in Section 3.2. A frozen DINOv2-L/14 encoder extracts slice features along the axial, coronal, and sagittal planes, and patch-aligned pooling reconstructs a cubic grid of local tokens. We use four transformer layers $(6, 12, 18, 24)$, each producing $16^3$ tokens per volume. Batch-based scoring is applied independently to each layer: tokens from layer $L$ across all volumes form a set of collections $\{C_i\}_{i=1}^B$ on which MSV-based scores are computed. We employ CoDe-Graph—referred to in this work as CoDeGraph3D—(Gia and Ahn, 2025), an extension of MuSc (Li et al., 2024) designed to handle consistent anomalies, using its default configuration. This yields four voxel-level anomaly maps that are averaged to obtain the final anomaly score. All per-axis tokens are projected to 128 dimensions using a fixed Gaussian random matrix to keep similarity computation tractable. Experiments are performed on a single NVIDIA RTX 4070 Ti Super GPU.

**Baseline Methods**   We compare our training-free framework with CLIP-based ZSAD and a representative reconstruction baseline. For CLIP-based models, we follow the protocol of Marzullo et al. (2025), evaluating AnomalyCLIP (Zhou et al., 2023) and APRIL-GAN (Chen et al., 2023). These methods operate at the slice level: each 2D slice is scored using a text-based anomaly measure, and slice-wise scores are aggregated across the volume to produce a 3D anomaly prediction. Unlike Marzullo et al. (2025), which considers only axial slices, we extend these methods to all three anatomical planes, aggregating their outputs (after interpolation to $224 \times 224$ resolution) to match the multi-view design of our approach. We use the official implementations of both models. The models are trained on an industrial AD dataset (MVTec (Bergmann et al., 2019)) at $224 \times 224$ resolution. Furthermore, we additionally trained CLIP-based methods on BraTS slices to provide a supervised reference for comparison. We also include WinCLIP (Jeong et al., 2023) as a purely zero-shot CLIP-based baseline, since it requires no fine-tuning. As the official implementation is not publicly available, we adapt an open-source PyTorch implementation[1] and extend it to 3D by applying the method in a slice-wise manner following the same protocol used for AnomalyCLIP and APRIL-GAN.

For unsupervised reconstruction, we implement a DAE (Kascenas et al., 2022), following the 3D-version configuration of Liang et al. (2026), using a standard 3D U-Net architecture (Xu et al., 2024). The model is trained exclusively on the IXI training subset. All details for training baseline methods are given in Appendix C.

---

[1] https://github.com/zqhang/Accurate-WinCLIP-pytorch

Table 1: **Quantitative comparison on T2-weighted MRI. Bold** indicates the best performance within each category (Zero-Shot vs. Reference).

| Method | Training Source | Patient-level | | Voxel-level | | | |
|---|---|---|---|---|---|---|---|
| | | AUROC | AP | AUROC | AP | Dice | IoU |
| *Zero-shot Methods (No medical training)* | | | | | | | |
| CoDeGraph3D | — | **96.9** | **92.7** | **92.2** | **49.1** | **41.3** | **31.6** |
| WinCLIP | — | 23.2 | 24.5 | 84.9 | 5.8 | 8.1 | 4.4 |
| AnomalyCLIP | Industrial | 36.4 | 28.6 | 91.2 | 11.8 | 14.1 | 8.4 |
| APRIL-GAN | Industrial | 3.5 | 21.2 | 90.0 | 8.7 | 12.7 | 7.3 |
| *Reference Methods (Requires domain training)* | | | | | | | |
| AnomalyCLIP | BraTS (Supervised) | 82.3 | 81.8 | 97.9 | 61.7 | 34.9 | 25.3 |
| APRIL-GAN | BraTS (Supervised) | 94.1 | 92.9 | **98.4** | **81.2** | **50.1** | **40.4** |
| DAE | IXI (Unsupervised) | **99.8** | **99.7** | 88.2 | 25.4 | 25.0 | 17.3 |

**Note:** *Training Source* indicates the dataset used for model optimization.

**Evaluation Metrics**  We follow standard evaluation practice in the anomaly detection community. Patient-level and voxel-level performance are assessed using AUROC and Average Precision (AP). Segmentation accuracy is additionally reported using Dice-max (Dice) and IoU, where Dice-max denotes the best Dice score obtained by thresholding the voxel-level anomaly map at the optimal threshold, following standard practice in unsupervised anomaly segmentation.

### 4.2. Results

**Comparison with Zero-Shot Baselines.**  On T2w (Table 1), CoDeGraph3D achieves a patient-level AUROC of **96.9%** and a voxel-level Dice of **41.3%**, validating that batch-based ZSAD is viable and effective for 3D brain MRI. Existing zero-shot baselines that rely on out-of-domain CLIP fine-tuning (AnomalyCLIP, APRIL-GAN) perform poorly in this setting, producing Dice scores below 15%. These results are consistent with the findings of Marzullo et al. (2025), who report similarly limited transfer of industrial CLIP models to 3D brain MRI. A similar trend is observed on T1-weighted MRI (Table 2), where CoDeGraph3D again substantially outperforms all zero-shot baselines. Beyond accuracy, the method remains efficient: processing all 180 volumes requires 714 seconds in total (4 seconds per volume) and uses less than 10GB VRAM (details in Table 8), confirming the practicality of our proposed method.

**Comparison with Reference Methods.**  To contextualize the zero-shot results, we additionally compare CoDeGraph3D with reference methods that require supervision. As expected, supervised CLIP-based models (AnomalyCLIP and APRIL-GAN trained on BraTS) achieve the highest AS performance. Meanwhile, CoDeGraph3D attains comparable seg-

Table 2: **Quantitative comparison on T1-weighted MRI. Bold** indicates the best performance within each category (Zero-Shot vs. Reference).

| Method | Training Source | Patient-level | | Voxel-level | | | |
|---|---|---|---|---|---|---|---|
| | | AUROC | AP | AUROC | AP | Dice | IoU |
| *Zero-shot Methods (No medical training)* | | | | | | | |
| CoDeGraph3D | — | **97.5** | **94.1** | **90.2** | **33.8** | **29.5** | **20.8** |
| WinCLIP | — | 34.6 | 38.1 | 70.0 | 2.6 | 4.4 | 2.3 |
| AnomalyCLIP | Industrial | 75.1 | 66.7 | 84.4 | 6.6 | 7.7 | 4.2 |
| APRIL-GAN | Industrial | 19.1 | 26.5 | 79.2 | 3.8 | 6.4 | 3.4 |
| *Reference Methods (Requires domain training)* | | | | | | | |
| AnomalyCLIP | BraTS (Supervised) | 84.8 | 84.7 | 96.7 | 52.5 | 30.4 | 21.0 |
| APRIL-GAN | BraTS (Supervised) | 80.9 | 81.8 | **97.8** | **74.8** | **42.1** | **32.5** |
| DAE | IXI (Unsupervised) | **100.0** | **100.0** | 69.1 | 4.3 | 7.6 | 4.3 |

**Note:** *Training Source* indicates the dataset used for model optimization.

mentation accuracy without any further training beyond the frozen foundation model. When compared with the unsupervised DAE trained on IXI normals, CoDeGraph3D lags in AC performance but clearly outperforms it in AS accuracy across both modalities. These results highlight that CoDeGraph3D provides a strong and practical trade-off between performance and training cost in the AC/AS for 3D brain MRI.

**Sensitivity to Lesion Scale.** In CoDeGraph3D, each token $\mathbf{v}_{(x,y,z)}$ represents a cubic region of size $(p \cdot s)^3$, where $p = 14$ is the patch size and $s \approx 0.7\,mm$ is the voxel spacing after preprocessing. In our experimental setup, this corresponds to an effective cube of approximately $(9.75\,\text{mm})^3$ ($\approx 926\,\text{mm}^3$) in the physical world. Lesions substantially smaller than this scale may therefore be partially attenuated by spatial averaging. Nevertheless, as shown in Lemma 1, small but sufficiently distinct anomalies can still remain detectable at the token level. This behavior is confirmed by the lesion-wise true positive rate (LTPR) analysis on BraTS-2025 METS. For lesions with volume $< 100\,\text{mm}^3$ (146/288 of all lesions)—which occupies roughly 10% of the effective cube volume—a non-negligible fraction can still be localized (LTPR = 0.23). In contrast, detection becomes more reliable for lesions larger than the effective cube size (LTPR = 0.83 for volumes $> 1000\,\text{mm}^3$).

**Qualitative Analysis.** Figure 1 presents representative anomaly segmentation results. CoDeGraph3D produces spatially coherent and well-localized anomaly maps, effectively avoiding the incoherent noise characteristic of slice-wise baselines. The method successfully delineates the primary extent of lesions with clear contrast against healthy tissue, validating the effectiveness of batch-based ZSAD. However, as expected, sensitivity is reduced in scenarios involving very small, scattered lesions (e.g., punctate metastases), where the coarse resolution of the cubic patch representation may dilute the anomaly signal.

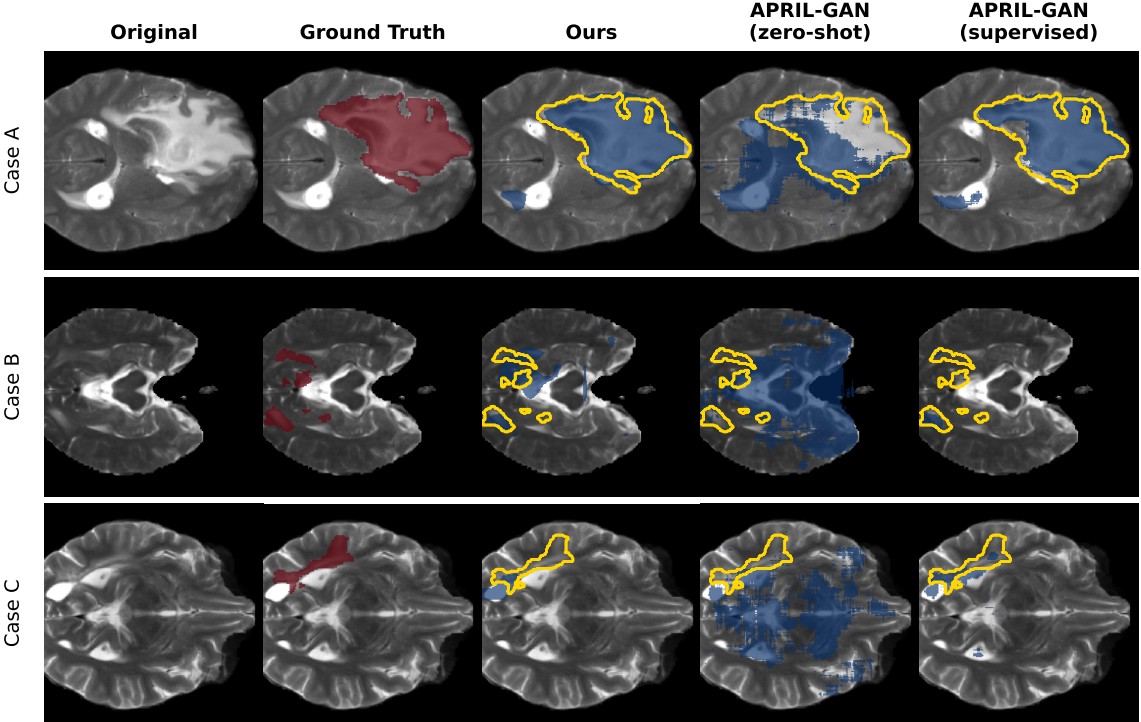

Figure 1: **Anomaly segmentation on 3D T2-weighted BraTS scans.** Yellow contours show ground-truth boundaries, and blue regions denote predicted anomaly masks. Rows illustrate high-, medium-, and low-Dice cases for CoDeGraph3D and APRIL-GAN baselines.

## 5. Ablation Studies

We quantitatively verify our tokenization strategy by testing it against alternative approaches. For all ablations, we report the results on the T2w modality of BraTS-2025 METS to assess their impact unless specified.

### 5.1. Reliability of Random Projections

We assess the sensitivity of our approach to the target dimensionality $k$ of the random projection $\mathbf{R}$ in Equation (2). Specifically, we test $k \in \{1, 10, 50, 100, 128, 200\}$ for three different seeds (Table 3). As expected, extremely low dimensions ($k \leq 10$) result in poor segmentation accuracy and higher variance, as the projection fails to preserve the pairwise distances required for anomaly detection. However, performance stabilizes after $k = 50$, with negligible variation observed between $k = 128$ and $k = 200$ (std of Dice $< 0.2\%$). These results indicate that aggressive dimensionality reduction (an $8\times$ reduction from $D = 1024$) is possible with only a marginal impact on stability, making the approach suitable for resource-constrained settings. This compression is critical for making quadratic batch-based comparisons computationally tractable.

Table 3: **Effect of Random Projection ($k$).** Performance stabilizes at $k \geq 50$. Values are mean $\pm$ std over 3 seeds.

| Method | $k$ | P-AUC | Dice |
|---|---|---|---|
| **CoDe Graph3D** | 1 | $71.4 \pm 6.8$ | $9.3 \pm 1.2$ |
| | 10 | $94.1 \pm 2.2$ | $25.9 \pm 1.0$ |
| | 50 | $96.4 \pm 0.5$ | $39.3 \pm 0.3$ |
| | 100 | $96.4 \pm 0.3$ | $40.6 \pm 0.4$ |
| | 128 | $96.9 \pm 0.3$ | $41.3 \pm 0.3$ |
| | 200 | $96.7 \pm 0.1$ | $41.5 \pm 0.1$ |

**Note.** P-AUC denotes patient-level AUROC.

Table 4: **Effect of Multi-View Aggregation.** Comparison of single-view and dual-view configurations.

| Method | Viewpoint | P-AUC | Dice |
|---|---|---|---|
| **CoDe Graph3D** | A | 98.0 | 36.9 |
| | C | 87.4 | 34.8 |
| | S | 99.4 | 23.5 |
| | C + S | 97.2 | 33.4 |
| | A + S | 98.0 | 39.6 |
| | A + C | 97.5 | 41.1 |
| | A + C + S | 96.9 | 41.3 |

**Note.** P-AUC denotes patient-level AUROC.

## 5.2. Importance of Multi-View Context

To assess the importance of multi-view processing, we compare models operating on single anatomical planes versus dual-view combinations (Table 4). We find that incorporating a second viewpoint consistently improves segmentation accuracy over single-view baselines. Among individual views, the Axial view alone yields the strongest individual performance (36.9% Dice), which may be attributed to the generally cleaner appearance of axial slices in BraTS-2025 METS, with less apparent blurring compared to other orientations. Notably, the dual-view (axial + coronal) configuration closely approaches the performance of the full tri-axial model (41.1% vs. 41.3%). This result suggests that while full 3D context is ideal, the aggregation of just two orthogonal planes already provides substantial geometric regularization against slice-wise inconsistency.

## 5.3. Robustness to Batch Size

We evaluate the sensitivity of CoDeGraph3D to the effective batch size used for batch-based comparisons by partitioning the test set into non-overlapping chunks of varying size $B$, with each chunk processed independently. As the batch size decreases, performance degrades gradually rather than abruptly. For moderate batch sizes ($B \geq 30$), both patient-level AUROC and voxel-level Dice remain stable and close to the full-batch setting, demonstrating that the method can be applied sequentially to large datasets without a significant loss in accuracy.

Notably, CoDeGraph3D remains functional even at even smaller batch sizes ($B = 15$), achieving non-trivial segmentation performance (Dice $\approx 37.5\%$), albeit with increased variance. These results indicate that the proposed method does not critically depend on large batch aggregation and can operate effectively under memory-constrained settings or in sparse-data regimes, where only a limited number of test samples are available.

Table 5: **Performance Stability with Chunking.** Patient-level AUROC and voxel-level Dice when the full test set is divided into random non-overlapping chunks of varying sizes.

| Number of Chunks | Batch Size ($B$) | Patient AUROC | Dice |
|:---:|:---:|:---:|:---:|
| 1 (Full Batch) | 180 | **96.9** | **41.3** |
| 2 | 90 | 96.7 | 40.8 |
| 4 | 45 | 96.4 | 40.0 |
| 6 | 30 | 96.0 | 39.8 |
| 12 | 15 | 95.2 | 37.5 |

Table 6: **Generalization to Glioma and Stroke.** Evaluation on BraTS-2021 GLI (T2w) and ATLAS R2.0 (T1w) against 115 IXI normals. *Baselines:* AnomalyCLIP/APRIL-GAN are fine-tuned on MVTec (Industrial); DAE is trained on IXI (Medical-Unsupervised).

| Method | BraTS-2021 GLI (Tumor) | | | | ATLAS R2.0 (Stroke) | | | |
|:---|:---:|:---:|:---:|:---:|:---:|:---:|:---:|:---:|
| | P-AUROC | P-AP | V-AP | Dice | P-AUROC | P-AP | V-AP | Dice |
| **CoDeGraph3D** | **99.4** | **98.9** | **51.6** | **61.0** | 87.2 | 72.6 | **34.4** | **31.6** |
| AnomalyCLIP | 68.1 | 57.4 | 17.1 | 22.5 | 53.7 | 47.2 | 4.1 | 4.8 |
| APRIL-GAN | 26.4 | 31.1 | 13.6 | 18.8 | 17.9 | 23.4 | 4.0 | 6.0 |
| DAE | 99.2 | 98.5 | 40.7 | 49.1 | **98.5** | **93.9** | 9.6 | 13.1 |

### 5.4. Ablation on Additional Anomaly Types

To assess the framework's generality, we conduct experiments on gliomas (BraTS-2021 GLI, T2w (Baid et al., 2021; Menze et al., 2014; Bakas et al., 2017)) and stroke lesions (ATLAS R2.0, T1w (Liew et al., 2022)). We keep the same 115 IXI normal scans as in the main setting and replace the 65 BraTS-2025 METS cases with 65 randomly sampled volumes from the target dataset. The preprocessing pipeline is identical to the main experiment, except that for ATLAS R2.0, IXI volumes are registered to MNI152 instead of SRI-24 to match ATLAS R2.0 registration. As shown in Table 6, CoDeGraph3D consistently achieves the highest segmentation accuracy across both diffuse gliomas (61.0% Dice) and vascular stroke lesions (31.6% Dice). These results confirm that the proposed batch-based 3D representation effectively generalizes to diverse anomalies.

## 6. Discussion and Conclusion

**Limitations.** A central limitation of CoDeGraph3D arises from its cubic tokenization strategy. Aggregating features over fixed spatial regions is essential for restoring 3D context and enabling tractable batch-based similarity computation, but it inherently constrains localization granularity. Consequently, very small, sparse, or low-contrast lesions may be attenuated by surrounding healthy tissue during feature aggregation, potentially leading to reduced sensitivity. In addition, although projection- and aggregation-based tokenization substantially reduces the dimensionality and number of features involved in similarity

computation, the underlying cross-sample similarity calculation still scales quadratically with the number of samples and tokens, which may limit applicability to extremely high-resolution volumes or very large test datasets.

**Future Directions.** Future work will focus on extending the proposed framework to achieve finer localization and improved scalability. This includes incorporating multi-scale or multi-resolution tokenization to better capture both large and small abnormalities, improving the efficiency of similarity computation to support larger cohorts and higher-resolution volumes. These extensions aim to further enhance the accuracy, efficiency, and practical applicability of training-free batch-based ZSAD in volumetric medical imaging.

Overall, this paper demonstrates that batch-based zero-shot anomaly detection can be effectively extended to 3D brain MRI without task-specific training, prompts, or domain adaptation. By constructing multi-axis volumetric tokens from frozen 2D foundation models, CoDeGraph3D enables rarity-based anomaly detection to operate in volumetric settings where slice-wise pipelines and text-driven ZSAD methods struggle. Across multiple modalities and anomaly types, the proposed framework consistently outperforms existing zero-shot baselines and achieves competitive voxel-level segmentation accuracy relative to supervised references, establishing CoDeGraph3D as a practical and domain-agnostic framework for training-free anomaly detection in 3D medical imaging.

## Acknowledgments

This research was supported by Basic Science Research Program through the National Research Foundation of Korea (NRF) funded by the Ministry of Education (No. RS-2023-00244515).

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

## Appendix A. Using Alternate 2D ViT Encoders

To demonstrate that CoDeGraph3D is adaptable to different foundation models, we replaced the default DINOv2 encoder with the CLIP visual encoder (ViT-L/14@336px) while keeping all other hyperparameters fixed.

Table 7: **Effect of Backbone Choice on BraTS T2-weighted.** Patient-level AUROC and voxel-level Dice of CoDeGraph3D using different 2D ViT encoders.

| Backbone | Patient AUROC (%) | Dice (%) |
|---|---|---|
| CLIP | 95.5 | 36.6 |
| DINOv2 | 96.9 | 41.3 |

As shown in Table 7, the framework remains effective with CLIP, achieving a Dice score of 36.6%—significantly higher than the slice-wise CLIP baselines reported in the main text. Table 7 also aligns with Gia and Ahn (2025), El Banani et al. (2024) and Oquab et al. (2023), which determined that DINOv2 yields more informative local features for general-purpose downstream tasks.

## Appendix B. Computational Efficiency

We evaluate the runtime and memory consumption of CoDeGraph3D across a range of batch sizes (Table 8). All experiments were conducted on a single NVIDIA RTX 4070 Ti Super GPU (16GB VRAM) using the default setting in Section 4. Unlike 2D batch-based ZSAD pipelines, where pairwise similarity computation with complexity $O(B^2 N^2 k)$ is often the primary bottleneck, our batch-based 3D formulation shifts most of the computational burden to the volumetric token construction step. This occurs because constructing volumetric tokens requires applying the 2D backbone to every slice along all three anatomical axes, while our design deliberately constrains both the number of 3D tokens per volume $N$ and the projected feature dimension $k$.

Table 8: **Computational Cost Analysis.** Breakdown of runtime and memory usage for increasing batch sizes. *Extraction* includes encoder forward pass and 3D tokenization; *Scoring* includes random projection, batch-graph construction, and voxel-level anomaly scoring.

| Batch Size (B) | Extraction (s) | Scoring (s) | Total Time (s) | Avg Time (s / vol) | Peak VRAM (GB) |
|---|---|---|---|---|---|
| 30 | 84.9 | 14.2 | 99.1 | 3.30 | 7.16 |
| 60 | 169.5 | 37.6 | 207.1 | 3.45 | 7.16 |
| 90 | 254.4 | 73.8 | 328.1 | 3.65 | 7.69 |
| 120 | 339.2 | 111.6 | 450.9 | 3.76 | 7.86 |
| 150 | 423.2 | 150.4 | 573.5 | 3.82 | 8.72 |
| 180 | 506.6 | 207.8 | 714.4 | 3.97 | 9.69 |

Peak memory usage remains below 10GB for all settings, confirming that CoDeGraph3D can be executed on widely available hardware. The per-volume runtime varies only modestly across batch sizes, indicating that the scoring component—despite its $O(B^2 N^2 k)$ complexity—does not impose significant overhead relative to the cost of constructing 3D patch tokens.

## Appendix C. Implementation Details of Baseline Methods

### C.1. 2D CLIP-Based Methods

In this appendix, we describe the configurations used for WinCLIP (Jeong et al., 2023), AnomalyCLIP (Zhou et al., 2023) and APRIL-GAN (Chen et al., 2023). For WinCLIP, we strictly follow the official settings (Jeong et al., 2023), employing the ViT-B/16+ backbone with an input resolution of $240 \times 240$ and the standard prompt ensemble. For Anoma-lyCLIP and APRIL-GAN, we use a frozen ViT-L/14-336 CLIP encoder, pretrained by OpenAI. Only the adapter layers are optimized. For the industrial setting, we train new checkpoints on MVTec-AD at $224 \times 224$ resolution to match our preprocessing pipeline. For the BraTS supervised setting, we apply the same preprocessing steps used in our main experiments (Section 4), extract axial, coronal, and sagittal slices, and retain at most ten tumor-containing slices per axis per volume (yielding a total of 7,819 slices from 263 BraTS training subjects).

For patient-level anomaly classification, voxel- or slice-level anomaly scores must be aggregated into a single subject-level score. For fine-tunable CLIP-based methods (Anoma-lyCLIP and APRIL-GAN), we empirically observe that using the maximum value of the predicted 3D anomaly map yields the most stable and strongest AC performance. Therefore, to ensure consistency across all CLIP-based baselines, we report AC metrics using voxel-wise max aggregation in the main paper. For WinCLIP, it operates without pixel-level fine-tuning and relies on CLIP representations optimized for image-level classification. As a result, aggregating slice-wise anomaly scores provides a better patient-level score. For transparency, we additionally report WinCLIP patient-level results using slice-wise aggregation in Table 10.

Table 9: Training configuration for CLIP-based baselines.

| Setting | AnomalyCLIP | APRIL-GAN |
|---|---|---|
| Backbone | ViT-L-14-336 | ViT-L-14-336 |
| Input size | $224 \times 224$ | $224 \times 224$ |
| Batch size | 8 | 8 |
| Epochs | 15 | 15 |
| Learning rate | 0.001 | 0.001 |
| Feature layers | [24] | [6, 12, 18, 24] |

Table 10: **WinCLIP Patient-Level Performance with Slice-wise Aggregation.** Patient-level anomaly classification results obtained by aggregating slice-wise CLS-token anomaly scores (max over slices).

| Dataset | AUROC | AP |
|---|---|---|
| BraTS-2025 METS (T1w) | 86.9 | 78.7 |
| BraTS-2025 METS (T2w) | 58.4 | 42.8 |
| BraTS-2021 GLI (T2w) | 75.4 | 68.8 |
| ATLAS R2.0 (T1w) | 62.5 | 51.5 |

DISCUSSION OF THE PERFORMANCE OF CLIP-BASED.

We acknowledge that our reported performance for CLIP-based models trained on BraTS is higher than the values reported in Marzullo et al. (2025), although the two studies agree on the poor performance of models fine-tuned on industrial datasets. With the exception of potential implementation differences—which we cannot assess because the authors did not release their training code—we suspect that the primary source of discrepancy lies in the sampling strategy. In particular, our implementation trains only on tumor-containing axial slices and already achieves strong performance (patient-level AUROC 86.3%, Dice 48.4%) even before incorporating coronal or sagittal views. As CLIP-based methods are not the focus of this work, these supervised results are included only as reference points to contextualize the performance of our training-free batch-based ZSAD framework.

### C.2. 3D Denoising Autoencoder

The DAE baseline follows the 3D configuration of Liang et al. (2026), implemented as a 3D U-Net with skip connections following Xu et al. (2024). Each input volume is centrally cropped to a cube of size $160^3$ and then Z-normalized. Training is performed using only the IXI healthy subset (462 samples). The main hyperparameters are summarized in Table 11.

Table 11: Configuration of the 3D DAE baseline.

| Setting | Value |
|---|---|
| Input crop size | $160^3$ |
| Voxel spacing | $1.0\,\text{mm}$ isotropic |
| Z-normalization | Yes |
| Noise level | Gaussian, $\sigma = 3.0$ |
| Noise resolution | $20 \times 20 \times 20$ |
| Epochs | 100 |
| Batch size | 2 |
| Optimizer | Adam |
| Learning rate | $1 \times 10^{-3}$ |
| Loss | L2 reconstruction |

## Appendix D. Patch-Level Sensitivity of 3D Tokens to Local Anomalies

In this section, we provide a simple formal argument showing that the proposed 3D patch tokens remain sensitive to sufficiently distinctive local anomalies, even when the anomaly occupies only a small fraction of a patch. To simplify the exposition, we focus solely on the effect of patch-level averaging and temporarily ignore feature normalization and random projection.

We consider a single anatomical view and a single transformer layer, and omit axis and layer superscripts for clarity. Along the depth direction, a cubic patch consists of $p$ slices, indexed by $\mathcal{P} = \{1, \ldots, p\}$. The corresponding patch token is obtained by averaging frozen encoder features $\{\mathbf{u}_t(\mathbf{x}) \in \mathbb{R}^D : t \in \mathcal{P}\}$:

$$\mathbf{z}_P(\mathbf{x}) = \frac{1}{p} \sum_{t \in \mathcal{P}} \mathbf{u}_t(\mathbf{x}) \; \in \; \mathbb{R}^D. \tag{4}$$

We consider two volumes $\mathbf{x}^N$ (normal) and $\mathbf{x}^A$ (anomalous). Let $\mathcal{A} \subset \mathcal{P}$ denote the subset of slices affected by the anomaly.

**Lemma 1 (Patch-Level Sensitivity to Local Anomalies)** *Let $\mathbf{x}^N$ and $\mathbf{x}^A$ be two volumes that differ only within a patch of $p$ slices, and let $\mathcal{A} \subset \mathcal{P}$ denote the anomalous subset with fraction $\alpha = |\mathcal{A}|/p$. Assume the encoder features satisfy:*

*(i) The average slice-wise feature difference over the anomalous slices is lower bounded:*

$$\left\| \frac{1}{|\mathcal{A}|} \sum_{t \in \mathcal{A}} \left( \mathbf{u}_t(\mathbf{x}^A) - \mathbf{u}_t(\mathbf{x}^N) \right) \right\| \; \geq \; \Delta_0 \tag{5}$$

*for some $\Delta_0 > 0$.*

*(ii) The feature difference outside the anomalous region is uniformly bounded:*

$$\left\| \mathbf{u}_t(\mathbf{x}^A) - \mathbf{u}_t(\mathbf{x}^N) \right\| \; \leq \; \varepsilon, \qquad \forall\, t \in \mathcal{P} \setminus \mathcal{A}, \tag{6}$$

*for some $\varepsilon \geq 0$.*

*Then the patch-level feature difference satisfies*

$$\left\| \mathbf{z}_P(\mathbf{x}^A) - \mathbf{z}_P(\mathbf{x}^N) \right\| \; \geq \; \alpha\, \Delta_0 \; - \; (1 - \alpha)\, \varepsilon. \tag{7}$$

*In particular, if $\alpha \Delta_0 >> (1 - \alpha)\varepsilon$, the anomalous patch token is strictly separated from the normal one.*

**Proof** Define $\Delta_t = \mathbf{u}_t(\mathbf{x}^A) - \mathbf{u}_t(\mathbf{x}^N)$. By linearity,

$$\mathbf{z}_P(\mathbf{x}^A) - \mathbf{z}_P(\mathbf{x}^N) = \frac{1}{p} \left( \sum_{t \in \mathcal{A}} \Delta_t + \sum_{t \in \mathcal{P} \setminus \mathcal{A}} \Delta_t \right). \tag{8}$$

Applying the triangle inequality yields

$$\left\| \mathbf{z}_P(\mathbf{x}^A) - \mathbf{z}_P(\mathbf{x}^N) \right\| \geq \frac{1}{p} \left\| \sum_{t \in \mathcal{A}} \Delta_t \right\| - \frac{1}{p} \left\| \sum_{t \in \mathcal{P} \setminus \mathcal{A}} \Delta_t \right\|. \tag{9}$$

For the anomalous term,

$$\left\| \sum_{t \in \mathcal{A}} \Delta_t \right\| = |\mathcal{A}| \left\| \frac{1}{|\mathcal{A}|} \sum_{t \in \mathcal{A}} \Delta_t \right\| \geq |\mathcal{A}| \, \Delta_0 = p\alpha\Delta_0,$$

by (5). For the complement,

$$\left\| \sum_{t \in \mathcal{P} \setminus \mathcal{A}} \Delta_t \right\| \leq \sum_{t \in \mathcal{P} \setminus \mathcal{A}} \|\Delta_t\| \leq (p - |\mathcal{A}|)\varepsilon = p(1 - \alpha)\varepsilon,$$

using (6). Substitution completes the proof. ∎

**Remark (Normalization and Random Projection).** In practice, patch features are $\ell_2$-normalized and optionally projected to a lower-dimensional space. On norm-bounded sets, the map $\mathbf{z} \mapsto \mathbf{z}/\|\mathbf{z}\|$ is Lipschitz, so the separation bound in Lemma 1 is preserved up to a constant factor after normalization. Furthermore, if a random projection $\mathbf{v}_P = \mathbf{R}^\top \mathbf{z}_P \in \mathbb{R}^k$ is applied, the Johnson–Lindenstrauss lemma guarantees that for any $0 < \vartheta < 1$, choosing $k = O(\vartheta^{-2} \log M)$ preserves all pairwise distances among $M$ patch tokens up to a factor $(1 \pm \vartheta)$ with high probability. Consequently, the scale-dependent separation established above is retained after both normalization and projection.

