# OpenReview forum: "Training-Free Zero-Shot Anomaly Detection in 3D Brain MRI with 2D Foundation Models"
_MIDL.io/2026/Conference — MIDL 2026 Poster_

### Official Review · Reviewer_fRWU · 2026-01-06

**Confidence:** 5
**Preliminary Rating:** 3
**Final Rating:** 4

**Summary:**

This work introduces a training-free framework for anomaly detection in 3D brain MRIs by using frozen 2D foundation (vision) models to extract features from volumetric data. The method aggregates these features into 3D tokens and identifies anomalies by calculating distances between tokens across a batch of images to find outliers. Experiments show that this method outperforms existing zero-shot baselines on brain tumor segmentation tasks while remaining computationally efficient enough to run on smaller hardware.

**Strengths:**

- The paper is overall well written and has a good structure. It is easy to follow. I also appreciate the details in the method, especially the equations and the patch tokenization method, which is well-explained.
- Overall, the motivations are interesting: there is indeed a need for more zero-shot AD methods, especially in 3D medical imaging.

**Weaknesses:**

- There is a huge limitation with the approach suggested by the authors: the huge downsampling factor to obtain the 3D vision tokens can make the framework totally miss smaller anomalies. Considering that the 3D patch tokenization is a main contribution of the framework, but also causes this huge limitation, I struggle to see how this approach would be successful in practical settings. Anomalies, by default, start as very small lesions (e.g., tumour or multiple sclerosis lesion).
- The comparison to the other zero-shot baselines that have seen industrial datasets during training (very different from 3D medical data) seems a bit unfair. More and/or more suitable baselines could have been selected instead.
- Performance is not as good as the authors make it seem. Also, it would have been more interesting to test this framework on different types of images and/or anomalies. The only result shown is for 3D brain tumours.
- See detailed comments for more.

**Detailed Comments:**

- There is a huge limitation with this method due to the 3D token downsampling. The authors also briefly point it out: smaller anomalies will likely be missed as false negatives. Can the authors discuss it in more detail, and how it may affect different types of lesions and anomalies? For instance, an analysis on the minimum voxel size an anomaly needs to be to be correctly represented in the 3D token and detected by the framework is needed (can then be converted to mm to give guidelines on the minimum size that can be detected). There are many cases of very small anomalies in brain MRI (e.g., multiple sclerosis lesions that are spread out, very small tumours that need to be identified early). The figure the authors present for the qualitative analysis is also biased, since it only shows examples with huge anomalies.
- In the results tables, the unsupervised DAE performs better than the authors' method on the patient-level metrics. Can the authors comment/discuss this?
- In terms of the comparison with zero-shot baselines, can the authors comment on the fairness of using two baselines that are trained on industrial AD (MVTec)? It seems to me that these two baselines performing worse than CoDeGraph would be expected here due to having seen industrial data. Would there have been better zero-shot baselines to use, or a fairer method/comparison?
- In the 3D patch tokenization method, could the authors have applied a different type of pooling (e.g., max instead of average)? What would have been the potential implications of this?
- The authors refer to CoDeGraph3D as the method here (CoDeGraph applied to 3D brain MRI), but in the tables, they still use CoDeGraph.

**Justification Of Final Rating:**

I appreciate the authors' willingness to improve the paper and clarify some claims and limitations. Although I still believe the tumour size detection is a considerable limitation, the authors provided a more in-depth analysis, and also added a more relevant baseline (WinCLIP). Therefore, I will increase my rating to a weak accept.

**Justification Of The Preliminary Rating:**

I am conflicted on this one. There is a major flaw in the method: it will likely miss small anomalies. Even though the authors mention this briefly, they do not discuss it. How can this affect different types/sizes of anomalies? Will this framework be usable in clinical settings even with this huge limitation? Also, the fairness in terms of baselines and datasets chosen is questionable.

I am willing to tend towards a weak accept if the authors (i) do an in-depth discussion on the risk of missing small anomalies and (ii) they evaluate their method on other types of datasets/anomalies to measure the limitations when working with smaller or different anomalies. This should be easy to do since the method is zero-shot.

**Questions To Address In The Rebuttal:**

The important things to address in the rebuttal are the following (see detailed comments above for more):
- The issue with this method likely missing small anomalies due to the downsampling in the 3D tokenization.
- The choice of baselines and/or potential to evaluate on other types of datasets/images/anomalies.

---

> ### Author Response · Authors · 2026-01-24
> **First part**
>
> We thank the reviewer for their thorough, technically insightful evaluation and for clearly specifying the conditions under which the work could be deemed acceptable. Below, we address the reviewer’s comments in a streamlined and consolidated manner, aligned with the main conceptual and technical concerns raised.
>
> ### **1) Sensitivity to small anomalies**
>
> > “There is a huge limitation with this method due to the 3D token downsampling, as smaller anomalies will likely be missed.”
>
> We agree that the coarse 3D tokenization imposes a fundamental limit on detecting very small anomalies. The revised manuscript now explicitly states the effective spatial resolution in physical units: under our experimental setup, each 3D token corresponds to a cubic region of approximately 9.75 mm per side (≈ 926 mm³). This gives a clear lower bound on the spatial scale at which anomalies can be reliably represented and indicates that lesions much smaller than this volume are likely to be diluted during aggregation.
>
> To quantify the practical consequence, we added a lesion-wise analysis on BraTS-2025 METS, which already includes a high proportion of small lesions (>50% lesions below 100 mm³). While very small lesions are frequently missed, detection is not impossible: roughly 20% of lesions <100 mm³ are still detected (i.e., LTPR ≈ 20%). For lesions larger than the effective cube size, CoDegraph3D achieves LTPR > 80%. These results align with the new theoretical analysis in Appendix D and confirm that the limitation, while significant, is not absolute.
>
> ### **2) Practical applicability**
>
> > The reviewer questions whether the framework can succeed in practical settings given the “huge limitation” that “smaller anomalies will likely be missed,” noting that anomalies often begin as very small lesions.
>
> The primary goal of our work is to provide a training-free, zero-shot anomaly segmentation framework that runs efficiently on consumer-grade GPUs for full 3D medical volumes. To make this feasible, we employ aggressive feature aggregation and coarse tokenization. These design choices sacrifice fine spatial resolution to prioritize computational efficiency and broad applicability (i.e., without any training or fine-tuning on medical domain datasets.) As a direct result, the method is biased to large anomalies and shows reduced detection performance to extremely small lesions—a known and expected trade-off, as already noted in both the original manuscript and the revision.
>
> Still, this limitation comes mostly from practical computing constraints, rather than  conceptual barriers. Finer resolution remains possible, though it would require significantly more memory and computation time. In the revised manuscript, we now explicitly discuss promising directions for future work—including more efficient similarity search, improved memory management, and hierarchical or multi-scale tokenization strategies—and include a clear caution that the current framework is not suitable for tasks dominated by very small lesions.
>
> ### **3) More baselines and more anomaly types**
>
> > The comparison with zero-shot baselines trained on industrial datasets may be unfair, and evaluation is limited to a single anomaly type.
>
> We appreciate this concern and have strengthened both the evaluation and its discussion. In the zero-shot anomaly detection (ZSAD) community, it is a common practice to fine-tune models on industrial datasets such as MVTec AD and then evaluate them on a wide range of domains—including medical images. Our original comparison follows this common protocol.
>
> That said, we agree that a stronger and fairer evaluation should also include a method that requires no fine-tuning on any external dataset (such as MVTec). To address this, we added WinCLIP—a purely zero-shot baseline that relies only on frozen foundation models and needs no training or adaptation at all—as a more direct reference point.
>
> To move beyond a single anomaly type, we also extended experiments to BraTS-2021 GLI (gliomas) and ATLAS R2.0 (stroke lesions), which differ markedly from metastases in morphology and etiology. The new results show that the framework generalizes reasonably well to other tumor types and non-tumor anomalies.

---

> ### Author Response · Authors · 2026-01-24
> **Second part**
>
> ### **4) Pooling choice**
>
> > Alternative pooling strategies (e.g., max pooling) could have been considered instead of average pooling.
>
> We chose average pooling for its simplicity and because it already delivers strong performance. Max pooling, by contrast, risks amplifying "rare" normal patterns or noise, which could distort the batch-level similarity statistics central to our approach. Exploring more sophisticated aggregation strategies remains a valuable direction for future research.
>
> ### 5) Patient-level performance of DAE
> > In the results tables, the unsupervised DAE performs better than the authors’ method on the patient-level metrics. Can the authors comment on this?
>
> DAE is trained on IXI and evaluated on BraTS-2025 METS, which introduces a  domain gap in terms of scanner characteristics, acquisition protocols, and pathology. Under this domain shift, accurate reconstruction of BraTS voxels becomes more difficult, leading to widespread reconstruction errors even in regions that are not strictly anomalous. Since patient-level scores in DAE are obtained by aggregating voxel-wise reconstruction errors, these global reconstruction difficulties can produce strong subject-level anomaly signals, resulting in high patient-level performance despite degraded voxel-level localization accuracy.
>
> ### 6) Other changes
>
> - We now consistently use “CoDegraph3D” (instead of “CoDeGraph”) when presenting results on 3D volumes.
> - We have replaced the main figure in the paper with a new one that better illustrates a wider range of lesion sizes.

---

### Official Review · Reviewer_aKFw · 2026-01-09

**Confidence:** 5
**Preliminary Rating:** 4
**Final Rating:** 4

**Summary:**

In this paper, the author propose to tackle the task of zero-shot anomaly detection (ZSAD), meaning anomaly detection without a training set (but with the test set sufficiently large so that it has enough normal samples), in 3D and medical images. They make the case that foundation visual (and language) models are only 2D, and thus to use their power for 3D medical imaging they need to  find a way to use 2D models for 3D volumes efficiently (the "amount of data" is much higher in 3D volumes and thus hard to scale). They propose to 1) work by 3D patch by summing the latent vector of 2D patches 2) perform a random projection to a much smaller subspace, with the intention of reducing the dimension and thus the computational load. They then use these "latent compressed cubes" with a batch-based anomaly detection method, CoDeGraph, already existing, by the same authors, to perform ZSAD on a mix of MRI from the IXI and BraTS datasets. They prove the superiority of their approach compared to ZSAD CLIP-based approaches (being fine-tuned on MVTecAD) for both classification and localization, and their superiority to a standard UAD approach for localization. CLIP-based models fine-tuned on BraTS still achieve superior results. They additionally show that the random projections work with a wide range of compression factors, and highlight the benefits of scanning the 3D volumes along multiple axes.

**Strengths:**

- The paper is well written, easy to follow
- The experiment design is clear
- The ablation studies are relevant
- The method proposed by the authors is quite simple and easy to reproduce
- Evaluation is performed on open databases

**Weaknesses:**

- The exact contribution in the paper (multi-view, patch aggregation and random projection) is actually modest, especially compared to the batch-based methods used or the foundation models
- The anomalies detected are very large (tumors) and the method is by design limited to detecting anomalies the size of the cubic patch
- The patch size, which is a very important parameter in this study, is not disclosed

**Detailed Comments:**

- The authors use the term " volumetric AC/AS" to say "volumetric anomaly classification and localization", I had to find this information in previous papers, please introduce these acronyms in the text.
- The authors state in the text that they will refer to their method as Codegraph3D but then in the results tables it's stated Codegraph
- segmentation accuracy is reported using Dice-max (Dice), what is "Dice-max" ? I did not find anything about "Dice-max" online, if there is any processing of the Dice metric please state it
- In the results table, the reviewer thinks it should be more honest either to have the best method across ZS and reference highlighted in bold, or better, the best method in ZS in bold *and* the best method in the reference methods in bold also.
- "However, as expected, sensitivity is reduced in scenarios involving very small, scattered lesions (e.g., punctate metastases), where the coarse resolution of the cubic patch representation may dilute the anomaly signal." This is a very interesting comment, the reviewer thinks it highlights the fact that the cubic patch size must be stated and discussed in the text. The very nature of working with cubic patches and then upsampling the score map is very limiting for small lesions detection. It might be worth mentioning previous works that tackle the issue of small/subtle lesion detection in AD.
- "These results suggest that the intrinsic anomaly manifold is low-rank and that embeddings can be aggressively compressed" This is a bit of an overstatement given the size and the contrast of the "anomalies" in the BraTS dataset
- Could the authors add A+S+C in table 3 for completeness ?
- It could be worth checking if the MRI aquisition in IXI or BraTS, for T1 and T2 was done in the axial (or else) axis, it could explain the better performance in this axis (less MR artifacts, better quality in this axis)
- The results presented in appendix, table 7 are very interesting and the reviewer think they could be integrated in the main body. See the comment just bellow but I think a reasonable and common question with batch-approaches is "what size should the batch be ? am I supposed to include the whole dataset ?"
- This point might be too meta or not the place for this but : the reviewer is genuinely wondering what differentiate "ZSAD batch methods" from regular UAD. ZSAD batch methods 1) either work with fundation models that have been trained on a very large corpus so they are not quite "zero shot" or 2) work with assumption that the "test" data contains a lot of normal samples and a small number of outliers. In this latter case, one can genuinely wonder if the "test set" is not indeed the training+testing set, as it is not like these methods could process a single sample for inference.

**Justification Of Final Rating:**

My original thoughts on the paper still stand :
-very well written
- experiments are well designed
- modest contribution
- clear limitation (by designed of the method)

However the authors have done a great work in replying and addressing the reviewers comments. I will confirm my weak accept rating.

**Justification Of The Preliminary Rating:**

The paper is very well written, the experiments are well designed. Despite the actual contribution being modest, and even though a clear limitation (by designed of the method) appears : the anomalies detected must be sufficiently large, the reviewer believes this work should be presented and discussed at the MIDL conference.

**Questions To Address In The Rebuttal:**

In general, see details comments but the curcial points here would be :
- Indicate the patch size very clearly
- Bold for both types of methods

---

> ### Author Response · Authors · 2026-01-24
> **First part**
>
> We thank the reviewer for the thorough, constructive, and well-balanced evaluation. Below, we address the reviewer’s comments in a streamlined and consolidated manner, aligned with the main conceptual and technical concerns raised.
>
> ---
>
> ### 1. Contribution, positioning, and scope of the paper
>
> > The individual components (multi-view aggregation, patch aggregation, random projection) are modest compared to existing batch-based methods and foundation models.
>
> The primary goal of our work is to deliver a practical, scalable framework for zero-shot anomaly detection and segmentation in 3D brain MRI that runs efficiently on consumer-grade GPUs. While the individual components are conceptually straightforward, our core contribution is showing that they can be combined in a principled way to enable fully training-free, dense 3D anomaly segmentation in realistic clinical settings.
>
> Our approach builds on the emerging line of research that adapts 2D foundation models to 3D medical tasks (e.g., RAPTOR [1], which successfully applied this strategy to classification and regression). We extend the idea to anomaly segmentation—a far more demanding task that requires strong spatial coherence, precise localization, and high computational efficiency under strict memory limits. We demonstrate that dense 3D anomaly segmentation is achievable without any 3D-specific pretraining or medical-domain fine-tuning, relying solely on frozen 2D foundation models. This goes well beyond a simple slice-wise application: true 3D segmentation demands spatially consistent and accurately localized representations despite tight resource constraints.
>
> ---
>
> ### 2. Patch size, spatial resolution, and sensitivity to small lesions
>
> > The patch (cube) size, which is a very important parameter in this study, is not disclosed
>
> We acknowledge (as already noted in the original submission) that the method is inherently limited by the size of the cubic patch representation. To address this, we have revised the manuscript to explicitly discuss and quantitatively characterize this limitation in a new subsection titled “Sensitivity to Lesion Scale” (Section 4.2). In this section, we clearly state that each 3D token corresponds to a cubic region of approximately (9.75 mm)³, or roughly 926 mm³, under the experimental settings used.
>
> To further evaluate the practical consequences of this design choice, we included a lesion-wise analysis on the BraTS-2025 METS dataset, along with a simple theoretical analysis in Appendix D. The results show that detection sensitivity decreases for very small lesions but remains non-negligible: for lesions smaller than 100 mm³ (approximately 10% of the effective cube volume), the lesion-wise true positive rate is around 20%. These empirical findings align with the theoretical analysis in Appendix D, which indicates that small but high-contrast anomalies can still be detected. For lesions larger than the effective cubic size, detection becomes markedly more reliable, with lesion-wise true positive rates exceeding 80% for lesions above 1000 mm³.
>
> ---
>
> ### 3. Batch Size and Inference Assumptions
>
> > The appendix on batch-size could be integrated in the main body; a common practical question is “how large should the batch be?” and whether inference implicitly assumes the whole dataset.
>
> Following your suggestion, we moved the batch-size analysis from the appendix to the main paper (now Section 5.3). In this section, the “chunking” experiment evaluates CoDeGraph3D under both computational constraints and limited sample availability. Specifically, we divide the test set into non-overlapping chunks and process each chunk independently—computing the Mutual Similarity Vector using only the volumes within that chunk. This setup loosely mimics a realistic clinical scenario: the full test set of 180 volumes represents all scans collected over one week, while each chunk corresponds to the scans acquired in a fixed time interval (e.g., daily basis). The results demonstrate that CoDeGraph3D performs effectively even when processing daily batches. In other words, the method does not require accumulating a full week’s data to generate reliable anomaly maps—it operates robustly with as few as 15 samples per batch and does not depend on access to the entire test dataset.
>
> [1] An, Ulzee, et al. "Raptor: Scalable train-free embeddings for 3d medical volumes leveraging pretrained 2d foundation models." _arXiv preprint arXiv:2507.08254_ (2025).

---

> ### Author Response · Authors · 2026-01-24
> **Second part**
>
> 4. Zero-shot definition and distinction from classical UAD
>
> In the AD literature, “zero-shot” usually refers to the absence of any optimization (e.g., fine-tuning) using data drawn from the target domain, rather than a complete absence of prior knowledge. In simple terms, zero-shot anomaly detection can be viewed as "transfer learning without optimization on the target dataset."
>
> Regarding the difference between classical UAD and batch-based ZSAD, the key distinction lies in the learning setting. Standard UAD is inductive: it relies on a separate, guaranteed-normal training set to learn a model that is later applied to individual test samples. In contrast, batch-based ZSAD methods are transductive. They assume that, within a given batch, normal patterns are common and anomalies are rare, and they detect anomalies by comparing samples against the statistical structure of the batch itself (similar to outlier detection methods such as LOF). The batch therefore does not serve as a training set for learning or updating the model, but only as a reference for identifying rarity. While batch-based ZSAD does not support single-sample inference, this reflects its transductive nature rather than a conflation of training and testing.
>
> ---
>
> ### 5. Others
>
> We addressed all presentation-related comments as follows:
>
> - Volumetric AC and AS are now explicitly defined as volumetric Anomaly Classification and Anomaly Segmentation at first use.
>
> - Dice-max is clarified as the maximum Dice score obtained by thresholding the voxel-level anomaly map, following standard practice in the AD field.
>
> - Result tables now clearly distinguish zero-shot methods and reference methods, with best results highlighted within each category.
>
> - The A + C + S config has been added to the multi-view ablation table.
>
> - We corrected the phrasing regarding low-rank structure to clarify that it applies to the normal manifold rather than anomalies: "anomaly manifold" -> "normal manifold"
>
> - Minor typographical and formatting issues were fixed throughout the manuscript.

---

> > ### Comment · Reviewer_aKFw · 2026-01-27
> >
> > The reviewer would like to thank the author for the added experiments, clarifications and discussion.
> >
> > There are still a few comments that the reviewer thinks were not addressed :
> >
> > - "These results suggest that the intrinsic normal manifold is low-rank and that embeddings can be aggressively compressed (an 8× reduction from D = 1024) with minimal trade-offs in stability" -> the reviewer still thinks this is an overstatement regarding the size and contrast of the BraTS dataset, this statement is more correct given the experiment with ATLASv2 but not known at this point in the text. The reviewer would still suggest to add nuance to this sentence.
> > - It could be worth checking if the MRI aquisition in IXI or BraTS, for T1 and T2 was done in the axial (or else) axis, it could explain the better performance in this axis (less MR artifacts, better quality in this axis)
> >
> >
> > Regarding the comments that the authors addressed :
> >
> > **1. Contribution, positioning, and scope of the paper** : the reviewer thanks the author for the discussion but still feel like the author just re-stated their goal and that the contribution is indeed modest, however given the quality of the experiments and the text, this is not an obstacle to being accepted at this conference from the reviewer's point of view
> > **2. Patch size, spatial resolution, and sensitivity to small lesions** : the reviewer thanks the authors for the added theoretical guarantee and the additional analysis. I would still affirm that 1) 100mm3 is quite big for a lesion 2) Talking about small lesions in term of volume proportion does not sound like the right way to analyze the situation, the authors could count the number of lesions greater/lower than a certain threshold (connected components) rather than their volume. This limitation on the size of the detectable lesions is a major weakness of the method.
> > **3. Batch Size and Inference Assumptions** : The reviewer is very convinced with the authors' answer.
> > **4.Zero-shot definition and distinction from classical UAD** : the reviewer would like to thank the authors for the quality of this discussion. I would still argue one last time for my point of view : you stated that ". The batch therefore does not serve as a training set for learning or updating the model, but only as a reference for identifying rarity", we could very much see here that "serving as a reference for identifying rarity" is exactly what a training set in UAD is supposed to do. Therefore there is indeed a kind of play around words as to what is ZSAD/UAD, what is/isn't the training set or the testing set.
> >
> > I will update my final rating just before the deadline but as of now I am 99% sure it will be weak accept or strong accept. Thank you for the great work.

---

> > ### Author Response · Authors · 2026-01-28
> >
> > We thank the reviewer for the careful follow-up and the positive assessment of the revised manuscript. Below, we address the two remaining points that were identified as not yet fully resolved.
> >
> > ### 1. Low-rank manifold statement.
> > We agree with the reviewer that experiments on BraTS alone are not sufficient to support a convincing claim about low-rank structure. We have therefore softened this statement and rewritten it in a purely empirical manner, focusing on the observed stability under dimensionality reduction rather than making claims about the intrinsic rank of the representation.
> >
> > The revised sentence now reads:
> > "These results indicate that aggressive dimensionality reduction (an $8\times$ reduction from $D=1024$) is possible with only a marginal impact on stability, making the approach suitable for resource-constrained settings."
> >
> > ### 2. Acquisition plane and axial performance.
> > We thank the reviewer for this suggestion. We attempted to determine the original MRI acquisition plane for both IXI and BraTS; however, we were unable to reliably recover this information from the released data or from the published descriptions of the BraTS dataset, which are provided as NIfTI volumes following DICOM-to-NIfTI conversion.
> >
> > As an alternative, we performed a qualitative inspection of the data and observed that, in BraTS, axial slices often appear visually cleaner, with less apparent blurring than other orientations (sagittal and coronal). We therefore revised Section 5.2 to include this observation as a possible contributing factor to the stronger performance observed on axial slices.
> >
> > The revised sentence now reads:
> > "While the axial view alone yields the strongest individual performance (36.9\% Dice), this may be attributed to the generally cleaner appearance of axial slices in BraTS-2025 METS, with less apparent blurring compared to other orientations."

---

> > > ### Comment · Reviewer_aKFw · 2026-01-29
> > >
> > > The reviewer is very satisfied with the authors responses.
> > >
> > > I will confirm my rating of weak accept and thank the authors for the work produced.

---

### Official Review · Reviewer_o1E8 · 2026-01-10

**Confidence:** 5
**Preliminary Rating:** 4
**Final Rating:** 4

**Summary:**

In this work, the authors propose to leverage 2D foundation models, such as DINOv2 to extract compact 3D representations of brain volumes in order to perform zero-shot anomaly detection. Their method consists in extracting slice-wise features along each of the 3 axis of the volume (axial, sagittal, coronal) and averaging the consecutive 2D representations to obtain a volumic representation. Then, the dimension of the representation is further reduced with a random Gaussian projection. Their method is evaluated on a test set composed of volumes from IXI (healthy) and BraTS (tumors) and compared to other zero-shot methods and unsupervised anomaly detection methods.

**Strengths:**

- Learning volumic representation from 3D (brain) images is a central problem in current research in medical imaging. The authors propose a novel and sound method to learn such a representation from pretrained 2D models, demonstrating that foundation models pre-trained on 2D natural images can be leveraged to analyze medical images.
- The paper is particularly well-written and easy to follow
- The ablation study is insightful to understand the importance of the main parameters of the method, in particular the number of random projections and the multi-view aggregation, although I think that one main experiment is missing from the ablation (please refer to next sections)

**Weaknesses:**

- The evaluation and comparison to existing baselines are limited. The methods are only compared on a task of tumor detection on BraTS, which is known to be a rather easy task, even for UAD methods. Only one unsupervised anomaly detection method (UAD), DAE, is implemented and it achieves much lower pixel-wise performance than reported in the literature [1]. See the detailed comments section for further details.
- Although ZSAD has significant advantages over methods that require a training phase, I feel like one of the major limitations of patch-based ZSAD is the size of the batch at inference. Indeed, with a smaller data size at inference, the variance of healthy and abnormal volumes might increase and make it more difficult for the model to distinguish between those two groups. The authors have included an experiment in the appendices where they increase the number of chunks during training, but to my understanding, the number of total volumes remains 180 and this experiment evaluates the robustness under memory limitations.

[1] Yu Cai, Weiwen Zhang, Hao Chen, Kwang-Ting Cheng, MedIAnomaly: A comparative study of anomaly detection in medical images, Medical Image Analysis, Volume 102, 2025.

**Detailed Comments:**

- The only UAD method the authors compared their CoDeGraph to is DAE. It is a strong baseline, but the UAD for medical imaging is much richer and I would like to see more methods in the evaluation.
- Moreover, I think the comparison is unfair towards the UAD frameworks. Indeed, DAE is trained on IXI which could lead to a domain gap with BraTS at the evaluation stage. In the literature [2], DAE have achieved much higher voxel-level scores in a 2D setup. In addition, this domain gap could explain why DAE is strong at a patient-level scale. As BraTS does not contain anomaly-free images, another way to compare ZSAD to UAD on BraTS would be to implement 2D versions of UAD models.
-  The evaluation is only done a single task, brain tumors detection and segmentation on BraTS. As stated by the authors, their method could be biased towards detecting large anomalies, hence explaining the good performance on BraTS. For completeness, additional experiments on other types of anomalies would be needed in my opinion. For instance white matter hyperintensities have also been used to evaluate anomaly detection methods.
- For reproducibility, I would encourage the authors to share the code associated with this work.

**Justification Of Final Rating:**

The authors have carefully responded to the comments of all the reviewers, which has led to a significant improvement of the paper quality. I would recommend a strong acceptance as the methodological novelty of the paper is rather limited, but I still believe that this work is a decent and valuable contribution to the field of ZSAD, which is why I am leaving my rating at weak accept.

**Justification Of The Preliminary Rating:**

The paper is well-written, clear and easy-to-follow. The topics of ZSAD and representation learning on 3D volumes are relevant for medical applications and the authors propose a sound approach to leverage 2D foundation models  Although the experiments could be improved by adding more tasks and more baselines, I would recommend accepting this work.

**Questions To Address In The Rebuttal:**

The authors are encouraged to respond to the questions addressed in the Weaknesses and Detailed Comments sections.

---

> ### Author Response · Authors · 2026-01-24
> **First part**
>
> We thank the reviewer for the positive assessment of our paper’s clarity and novelty, and for the constructive feedback regarding our baselines and evaluation scope. Below, we address the reviewer’s comments in a streamlined and consolidated manner, aligned with the main conceptual and technical concerns raised.
>
> ### 1. Limited Baselines and UAD Performance
>
> > Comparison is limited to DAE; DAE performance is lower than literature; Comparison is unfair due to domain gap (IXI vs. BraTS).
>
> We understand the reviewer’s concern regarding the limited evaluation of UAD baselines and the reported performance of DAE. We first clarify that several works (as you cited) reporting DAE performance on BraTS-2021 GLI (glioma) rather than BraTS-METS (metastases), which is used in our main experiments and presents different lesion characteristics. To enable a more direct comparison and to diversify anomaly types, we therefore added experiments on BraTS-2021 GLI (along with ATLAS R2.0) in the revised manuscript. As shown in Table 6, our 3D DAE implementation achieves a Dice score of 49.1%, which is consistent with results reported in the literature (e.g., IterMask3D). This confirms the correctness and competitiveness of our DAE implementation.
>
> We agree that the domain gap introduced by training on IXI and testing on BraTS largely explains the discrepancy between DAE’s strong patient-level anomaly classification performance and its weaker voxel-level localization accuracy. This behavior is expected for reconstruction-based models under domain shift: accurate voxel-level reconstruction becomes difficult when distributions differ, while large reconstruction errors still provide a strong signal for subject-level detection. Rather than viewing this setting as unfair, we consider it representative of realistic clinical deployment, where differences in scanners, protocols, or patient populations are common. In contrast, our training-free approach avoids domain-specific optimization and relies on representations learned by foundation models, which contributes to improved robustness under such shifts.
>
> ### 2. Robustness to Batch Size
> > *Batch-based approach depends on batch size; smaller batches increase variance. The chunking experiment only evaluates memory constraints.*
>
> We agree that our batch-based similarity estimation relies on sample statistics, and a smaller batch-size naturally increase variance, which can affect AC/AS performance. And following the suggestion from Reviewer aKFw, we have relocated the detailed batch-size analysis to a clearer location in Section 5.3 (Table 5) for better visibility.
>
> We would also like to clarify that the “chunking” experiment is not just a test of memory limits—it directly assesses statistical robustness when only a small number of samples are available at once. In this setup, the test set (180 volumes) is divided into non-overlapping chunks, and similarity statistics are computed independently within each chunk. This loosely mimics real-world clinical deployment where only a limited number of cases may be processed together at a fixed time-interval (e.g., daily acquisitions rather than a full week’s worth of scans). The results demonstrate that CoDeGraph3D remains usable even with quite small batches: performance declines smoothly as batch size decreases, but the method still achieves reasonable segmentation (Dice ≈ 37.5%) at **B = 15**. This indicates that large concurrent batches are helpful but not strictly required for effective operation.
>
> [1] Liang, Ziyun, et al. "IterMask3D: Unsupervised Anomaly Detection and Segmentation with Test-Time Iterative Mask Refinement in 3D Brain MR." _arXiv preprint arXiv:2504.04911_(2025).

---

> > ### Comment · Reviewer_o1E8 · 2026-02-01
> >
> > I would like to thank the authors for their detailed answers and the effort they put into responding to the comments. As said in my official rating, I have not improved my rating, but I do acknowledge and value the improved quality of the paper (more results, datasets, etc.)

---

> ### Author Response · Authors · 2026-01-24
> **Second part**
>
> ### 3. Evaluation on Additional Anomaly Types
>
> > _The method may be biased toward large anomalies; additional experiments on other types are needed._
>
> We agree that testing a wider variety of anomaly types is essential to demonstrate generalization. To address this, we added a new Section 5.4 and Table 6 with results on two additional datasets: BraTS-2021 GLI (gliomas, different tumors) and ATLAS R2.0 (ischemic stroke lesions, non-tumor anomalies). On these datasets, CoDeGraph3D consistently outperforms all zero-shot baselines as well as the unsupervised DAE, achieving 61.0% Dice on GLI and 31.6% Dice on ATLAS. These gains show that the framework generalizes effectively beyond a single tumor type.
>
> We also acknowledge the reviewer’s concern that—by design—the method favors anomalies with larger size. To better characterize this built-in bias, we added a paragraph in Section 4.2 on lesion-scale sensitivity, supported by a detailed lesion-wise analysis on BraTS-2025 METS. We detail the effective token volume (≈ 926 mm³), and we  quantitatively show that detection sensitivity drops for very small lesions (< 100 mm³), yet a meaningful fraction are still caught (lesion-wise TPR ≈ 20%). Performance improves markedly for lesions larger than effective cube size (LTPR > 80% once volume > 1000 mm³). These results align with the simplified theoretical analysis added in Appendix D, which indicate that the size-related limitation, while real and important, is not absolute.
>
> ### 4. Others
>
> - Code Publish: we are committed to reproducibility. We will release the full source code upon acceptance.
>
> We hope these additional experiments and clarifications address your concerns regarding the baselines and robustness of our method.

---

### Author Rebuttal · Authors · 2026-01-24

**Rebuttal:**

We have uploaded the revised manuscript. Major and minor revisions are listed below:

Major revisions

- Added WinCLIP as an additional fully training-free zero-shot baseline in the main experiments.
- Added a new subsection “Sensitivity to Lesion Scale” that clarifies the localization limits of cubic tokenization and reports lesion-size–stratified localization results.
- Added experiments on BraTS-2021 GLI (glioma) and ATLAS R2.0 (stroke) to demonstrate generality across anomaly types.
- Moved “Robustness to Batch Size” from the Appendix to the main paper and revised it for clearer interpretation.
- Added a simplified theoretical model to support the discussion of small-lesion detectability under patch-level averaging.
- Reorganized the manuscript structure by moving prior limitations into a dedicated Discussion and Conclusion section.
- Added a “Future Directions” subsection to Discussion and Conclusion.

Other updates

- Standardized terminology (use CoDeGraph3D consistently).
- Replaced the main qualitative figure with a new version that better illustrates a wider range of lesion sizes.
- Defined volumetric AC/AS at first use and clarified Dice-max.
- Applied bold formatting to both category headers in Table 1 and Table 2.
- Added the missing A+C+S entry in the multi-view ablation.
- Corrected the phrase about the low-rank claim: changed "anomaly manifold" to "normal manifold".
- Minor fixes: corrected typographical issues and formatting inconsistencies throughout.

**Supporting Material:**

/attachment/40c448220af964a41a906221768ba7724ec883ae.pdf

---

> ### Author Response · Authors · 2026-01-25
> **Additional small fixes**
>
> - Corrected an error in Table 2 for AnomalyCLIP on BraTS-2025 T1-weighted MRI due to an incorrect input in the original draft (voxel-level AUROC: 6.6 → 84.4, voxel-level AP: 7.7 → 6.6).
>
> - Softened the low-rank manifold claim by rephrasing it as an empirical observation on stability under dimensionality reduction, as suggested by reviewer aKFw.
>
> - Clarified the multi-view context analysis by noting a plausible explanation for the strong axial performance based on the generally cleaner appearance of axial slices, as suggested by reviewer aKFw.

---

### Comment · Area_Chair_D2Uw · 2026-01-29
**Final Rating**

Dear Reviewers,
we appreciate your active participation in the discussion.
Please consider the rebuttals of the authors, as well as the revised manuscripts, and set your final rating by clicking "Edit"->"Official Review" by  February 1st 2026 (23:59 AoE).
best regards

---

### Meta-Review · Area_Chair_D2Uw · 2026-02-03

**Recommendation:** Accept (Poster)
**Confidence:** 4

**Metareview:**

All reviewers agree that the paper is well written and easy to follow. However, there are some limitations especially regarding few comparing methods included (compared to the number of UAD methods in the literature) as well as the insensitivity to small lesions. However, all reviewers rate the work as a "weak accept" and I therefore recommend an "accept". I'd like to thank the reviewers and the authors for their work!

---

### Decision · Program_Chairs · 2026-02-14

Accept (Poster)